# A Parent Version of the Motors of COVID-19 Vaccination Acceptance Scale for Assessing Parents’ Motivation to Have Their Children Vaccinated

**DOI:** 10.3390/vaccines11071192

**Published:** 2023-07-03

**Authors:** Chung-Ying Lin, Ray C. Hsiao, Yu-Min Chen, Cheng-Fang Yen

**Affiliations:** 1Institute of Allied Health Sciences, College of Medicine, National Cheng Kung University, Tainan 70101, Taiwan; 2Department of Occupational Therapy, College of Medicine, National Cheng Kung University, Tainan 70101, Taiwan; 3Department of Public Health, National Cheng Kung University Hospital, College of Medicine, National Cheng Kung University, Tainan 70101, Taiwan; 4Biostatistics Consulting Center, National Cheng Kung University Hospital, College of Medicine, National Cheng Kung University, Tainan 70101, Taiwan; 5Department of Psychiatry, Seattle Children’s, Seattle, WA 98195, USA; 6Department of Psychiatry and Behavioral Sciences, School of Medicine, University of Washington, Seattle, WA 98105, USA; 7Department of Psychiatry, Kaohsiung Medical University Hospital, Kaohsiung 80756, Taiwan; 8Department of Psychiatry, School of Medicine College of Medicine, Kaohsiung Medical University, Kaohsiung 80708, Taiwan; 9College of Professional Studies, National Pingtung University of Science and Technology, Pingtung 91201, Taiwan

**Keywords:** child, COVID-19, motivation, parent, vaccine

## Abstract

Parents’ motivation to vaccinate their children against coronavirus disease 2019 (COVID-19) plays a crucial role in the uptake of COVID-19 vaccines among children. The Motors of COVID-19 Vaccination Acceptance Scale (MoVac-COVID19S) is a valuable tool for assessing individuals’ vaccination-related attitudes and the factors influencing their decision to be vaccinated against COVID-19. This study adapted the MoVac-COVID19S to create a parent version (P-MoVac-COVID19S) and examined the psychometric soundness of two P-MoVac-COVID19S versions (a 9-item version (P-MoVac-COVID19S-9) and a 12-item version (P-MoVac-COVID19S-12)) for assessing parents’ motivation to vaccinate their children. A total of 550 parents completed the P-MoVac-COVID19S and a questionnaire assessing the factors that impact parents’ intention to allow their children to receive the COVID-19 vaccine using a vaccine acceptance scale. We enquired about the level of parental worry regarding the adverse effects of COVID-19 vaccines on children’s health and the number of COVID-19 vaccine doses received by parents. The factor structures of the P-MoVac-COVID19S-9 and P-MoVac-COVID19S-12 were examined using confirmatory factor analysis. The internal consistency, test–retest reliability, and concurrent validity of the P-MoVac-COVID19S were also examined. The results revealed that the P-MoVac-COVID19S-12 has a four-factor structure, which aligns well with the theoretical framework of the cognitive model of empowerment; the P-MoVac-COVID19S-9 has a one-factor structure. Both the P-MoVac-COVID19S-9 and P-MoVac-COVID19S-12 had good internal consistency and test–retest reliability and acceptable concurrent validity. The results of this study demonstrated that the P-MoVac-COVID19S is a reliable and valid instrument for assessing parent’s motivation to vaccinate their children against COVID-19.

## 1. Introduction

The Director-General of the World Health Organization (WHO) announced that coronavirus disease 2019 (COVID-19) no longer constitutes a public health emergency of international concern on 3 May 2023 because of the decreasing numbers of COVID-19 deaths, the decline in COVID-19-related hospitalizations, and high levels of population immunity [1]. However, in April 2023, over 10,000 COVID-19 cases were reported in children per week in the United States [2]. Vaccination against COVID-19 is one strategy to prevent COVID-19 infection in children [3]. COVID-19 vaccines are now available for immunizing children over 6 months of age [3]. Vaccination against COVID-19 can reduce the risk of COVID-19 infection and hospitalization [4,5]. Studies have revealed that most parents are willing to vaccinate their children to reduce their risk of contracting COVID-19; however, many parents remain hesitant about vaccine uptake among children [6,7,8,9,10,11,12,13,14]. A systematic review and meta-analysis on 44 studies published between 2020 and 2021 found that 60.1% of parents intended to have their child vaccinated against COVID-19, 22.9% of parents had no intention to have their child vaccinated, and 25.8% of parents were unsure [15]. The predictors of parents’ intention to have their child vaccinated against COVID-19 included being fathers, being parents of older age, having higher income, having a higher level of perceived COVID-19 threat, and having positive attitudes towards vaccines and vaccination [15]. Another review study revealed that the leading reason for parents’ vaccinating their child against COVID-19 was to protect children and family members, while the most important reason for not having their child vaccinated was the fear of side effects and the safety of vaccines for children [16]. A review study demonstrated that the willingness of parents to vaccinate their children was influenced by public attitudes; a more positive public attitude towards vaccination correlated with higher vaccination rates [17]. Assessing parents’ attitudes and the factors affecting their choice to vaccinate their children and developing intervention programs to increase children’s rate of vaccination against COVID-19 are essential.

A valid instrument is essential to assess individuals’ attitudes and the factors affecting the decision to be vaccinated against COVID-19. There have been several instruments developed for measuring individuals’ attitudes toward COVID-19 vaccines such as the Arizona CoVHORT Vaccine Questionnaire [18]; the Adult Vaccine Hesitancy Scale [19]; the Oxford COVID-19 Vaccine Hesitancy Scale [20]; the COVID-19 Vaccine Attitudes and Beliefs Scale [21]; the COVID-19 Vaccine Concerns Scale [22]; the Multidimensional COVID-19 Vaccine Hesitancy Scale [23]; the COVID-19 Vaccine Hesitancy Questionnaire [24]; the Vaccine Hesitancy Scale on Knowledge, Attitude, Trust and Vaccination Environment [25]; the COVID-19 Vaccine Hesitancy Scale in Qatar [26]; and the 5C Scale [27]. The Motors of COVID-19 Vaccination Acceptance Scale (MoVac-COVID19S) is a valuable tool for assessing individuals’ attitudes and factors affecting their decision to be vaccinated against COVID-19 [28,29,30]. The MoVac-COVID19S has several advantages in measuring individuals’ attitudes toward COVID-19 vaccines. First, the MoVac-COVID19S was adapted from the Motors of Influenza Vaccination Acceptance Scale, which was developed for assessing acceptance of the influenza vaccine [31]. The MoVac-COVID19S, which is based on the cognitive model of empowerment (CME) [32], incorporates four core cognitive components that determine individuals’ motivation to be vaccinated against COVID-19, namely, values (i.e., how much an individual cares about the purpose of vaccination), impacts (i.e., how much an individual believes in the effectiveness of COVID-19 vaccination), knowledge (i.e., an individual’s level of knowledge regarding vaccination against COVID-19), and autonomy (i.e., an individual’s confidence and control over their decision to receive the COVID-19 vaccine). Compared with most other instruments, the MoVac-COVID19S assesses a broader scope of understanding the attitudes toward COVID-19 vaccines. Second, most other instruments have been validated among people living in a single country or region. Studies have verified that the MoVac-COVID19S has acceptable psychometric soundness for assessing individuals’ motivation to be vaccinated in different populations, including in Taiwan, mainland China, India, Ghana, Afghanistan, Indonesia, and Malaysia [33,34]. Third, two versions of the MoVac-COVID19S have been proposed: a 9-item MoVac-COVID19S with all items worded positively and a 12-item MoVac-COVID19S with 9 items worded positively and 3 items worded negatively [28]. The two versions of the MoVac-COVID19S offer a more flexible way of assessing individuals’ attitudes toward COVID-19 vaccines depending on the needs of the surveys. The summary of the instruments developed for assessing individuals’ attitudes toward being vaccinated against COVID-19 is shown in Table 1.

Whether the MoVac-COVID19S can be used to effectively assess parents’ motivation to have their child vaccinated against COVID-19 warrants examination. If the P-MoVac-COVID19S can be employed to effectively assess parents’ motivation to have their child vaccinated, the MoVac-COVID19S and P-MoVac-COVID19S can be simultaneously used to compare parents’ motivation to vaccinate themselves and their children. The present study aimed to create the parent version of the MoVac-COVID19S (P-MoVac-COVID19S), designing a 9-item version and a 12-item version, by modifying the MoVac-COVID19S and examining its psychometric propensities, including factor structures, test–retest reliability, and internal consistency. We designed the 12-item P-MoVac-COVID19S (the P-MoVac-COVID19S-12) to have a four-factor structure (values, impacts, knowledge, and autonomy) and the 9-item P-MoVac-COVID19S (the P-MoVac-COVID19S-9) to have a one-factor structure, corresponding well with the CME theory. We hypothesized that the P-MoVac-COVID19S-9 and P-MoVac-COVID19S-12 had acceptable internal consistency and test–retest reliability. According to protection motivation theory [35,36,37,38,39], we also studied whether the P-MoVac-COVID19S-9 and P-MoVac-COVID19S-12 had acceptable concurrent validity by analyzing the association between parent’s motivation to vaccinate their children and the level of parental worry regarding the adverse effects of COVID-19 vaccination on children’s health and the number of COVID-19 vaccine doses received by parents.

## 2. Methods

### 2.1. Participants and Procedure

This study enrolled parents who were aged 20 years or older and had a child aged between 6 and 18 years old. Participants were recruited using an online advertisement posted on social media platforms, such as Facebook, Twitter, LINE (a direct messaging app commonly used in Taiwan), and the PPT bulletin board system from August 2022 to April 2023. Interested parents were instructed to contact the research assistants, who ensured the eligibility of potential participants, explained the study aims and procedures, and scheduled a time for eligible participants to individually complete the study questionnaires in a quiet study room. In total, 562 parents expressed interest in participating in research; of them, 12 parents were excluded because of their children’s age (younger than 6 or older than 18 years). A total of 550 parents participated in the study. The potential participants were also assessed by the research assistants to determine whether they had signs of impaired intellect or substance use that might interfere with their understanding of the study’s purpose or prevent them from completing the questionnaire. No participants were excluded. Informed consent was obtained from all participants prior to the assessment. Participants completed the study questionnaire in an on-site study room. This study was approved by the Institutional Review Board of Kaohsiung Medical University Hospital (KMUHIRB- KMUHIRB-E(I)-20220107).

### 2.2. Measures

#### 2.2.1. P-MoVac-COVID19S

The research team adapted the MoVac-COVID19S into P-MoVac-COVID19S by replacing the words “me” and “my” with “my child” and “my child’s”, respectively. Item 7, “I feel pressured about receiving COVID-19 vaccine”, was transformed into “I feel pressured about letting my child receive COVID-19 vaccine”. Item 11, “I only receive COVID-19 vaccine if it is required”, was transformed into “I only let my child receive COVID-19 vaccine if it is required”. The P-MoVac-COVID19S-12 comprises four domains, each with four items. The domains are composed of values (sample item: Vaccinating my child against COVID-19 is important), impacts (sample item: Vaccination greatly reduces my child’s risk of COVID-19 infection), knowledge (sample item: I understand how the vaccine helps my child’s body fight the COVID-19 virus), and autonomy (sample item: I can choose whether to allow my child to be vaccinated against COVID-19 or not). All items are rated based on a 7-point Likert-type scale (1 = strongly disagree; 7 = strongly agree), and three items (i.e., items 7, 10, and 11) are reverse-coded to ensure that higher summed scores on the P-MoVac-COVID19S indicate higher levels of parental acceptance to vaccinate children against COVID-19. In addition to the P-MoVac-COVID19S-12, a P-MoVac-COVID19S-9 was developed. Both the 12-item and 9-item versions of the MoVac-COVID19S have been revealed to be valid and reliable [28,33]. However, the 9-item version had a better data–model fit than the 12-item version. Moreover, the 9-item MoVac-COVID19S had a better fit with a one-factor structure (compared with the four-factor structure of the cognitive model of empowerment), whereas the 12-item MoVac-COVID19S had a better fit with a four-factor structure (compared with a one-factor structure).

#### 2.2.2. Vaccination Intention and Level of Worry

Three items were used to evaluate information related to COVID-19 vaccination: (a) parents’ intention to allow their children to receive a COVID-19 vaccine (rated on a vaccination attitude scale from 1 to 10); (b) the level of parental worry regarding the adverse effects of COVID-19 vaccination on children’s health (rated based on a 5-point Likert scale from not at all worried to extremely worried); and (c) the number of COVID-19 vaccine doses received by parents (parents were asked to fill in a number).

#### 2.2.3. Demographic Characteristics

Studies have found that parents’ motivation to vaccinate their child varied across different sex and ages of parents and children [13,40,41]; therefore, data on the sex and ages of the parents and their children were collected. Moreover, parental education level was a key factor influencing pediatric COVID-19 vaccine hesitancy [40]; therefore, the parents were asked to state how many years of education they had received.

### 2.3. Data Analysis

Descriptive statistics were used to summarize the characteristics of the study sample and the properties of the P-MoVac-COVID19S items. Spearman’s rank correlation coefficients were employed to evaluate the test–retest reliability (with a time interval of one week) of each item on the P-MoVac-COVID19S and of the two summed scores (i.e., P-MoVac-COVID19S-12 and P-MoVac-COVID19S-9). When the correlation coefficient is larger than 0.4, the test–retest reliability is considered to be satisfactory; when the coefficient is between 0.3 and 0.5, the reliability is acceptable [42]. Confirmatory factor analysis (CFA) with diagonally weighted least squares was applied to analyze the two versions of P-MoVac-COVID19S with two factor structures (i.e., each version was tested using a four-factor structure based on the cognitive model of empowerment and using a one-factor structure). The diagonally weighted least squares estimator was used because it can handle an ordinal scale, such as the Likert-type scales used in the P-MoVac-COVID19S [43]. Fit statistics were employed to evaluate the data–model fit, namely, the comparative fit index (CFI), the Tucker–Lewis index (TLI), the root-mean-square error of approximation (RMSEA), and the standardized root-mean-square residual (SRMR). Fit statistics were indicated when the CFI and TLI values were >0.9 and the RMSEA and SRMR values were <0.08 [43]. Moreover, the factor structures of the P-MoVac-COVID19S-12 and P-MoVac-COVID19S-9 were separately compared using *χ*^2^ tests. The internal consistency and concurrent validity of both the P-MoVac-COVID19S-12 and P-MoVac-COVID19S-9 were also examined. Internal consistency was evaluated using Cronbach’s α, with a value of >0.7 indicating good internal consistency [44]. Concurrent validity was evaluated by examining the association of three external criteria (i.e., intention of parents to allow their children to be vaccinated; level of worry about adverse effects of vaccination on children; number of vaccine doses received by parents) with P-MoVac-COVID19S-12 and P-MoVac-COVID19S-9 scores. All statistical analyses were conducted using SPSS 20.0 (IBM, Armonk, NY, US), except for the CFA, which used the lavaan R package [45].

## 3. Results

The parent sample had an average age of 44.29 (SD = 5.24) years, and the child sample had an average age of 11.80 (SD = 3.57) years, as presented in Table 2. Over three quarters of the parent sample were women (*n* = 427; 77.6%); the sex distribution in the child sample was relatively balanced (*n* = 301; 54.7% male). The parents were relatively well educated, with an average of 16.25 years of education (SD = 2.47). Most of the parents had received three or more doses of the COVID-19 vaccine (*n* = 513; 93.3%). The parents tended to be willing to allow their children to be vaccinated against COVID-19 (mean of intention = 7.87 on a vaccination attitude scale of 1–10), and almost half the parents had a moderate or high level of worry regarding the adverse effects of vaccination (*n* = 273; 49.6%).

The properties of the P-MoVac-COVID19S items are provided in Table 3. In brief, the mean values of the twelve items were relatively high (3.74 to 5.86 on a 7-point Likert-type scale). Moreover, all items were distributed normally (skewness = −1.694 to 0.022; kurtosis = −1.133 to 4.080) and had relatively good test–retest reliability (*r* = 0.39 to 0.65; all *p* < 0.001).

Regarding the factor structure of the P-MoVac-COVID19S (Table 4), the P-MoVac-COVID19S-12 had better fit with a four-factor structure (CFI = 0.945, TLI = 0.924, RMSEA = 0.084, and SRMR = 0.088) than with a one-factor structure (CFI = 0.933, TLI = 0.918, RMSEA = 0.088, and SRMR = 0.095), with a significant *χ*^2^ difference between the two structures (Δ*χ*^2^ = 47.31, Δ*df* = 6; *p* < 0.001). The P-MoVac-COVID19S-9 also had better fit with a four-factor structure (CFI = 0.997, TLI = 0.995, RMSEA = 0.026, and SRMR = 0.050) than with a one-factor structure (CFI = 0.995, TLI = 0.993, RMSEA = 0.030, and SRMR = 0.059); however, the *χ*^2^ difference between the two factor structures was nonsignificant (Δ*χ*^2^ = 10.33, Δ*df* = 5; *p* = 0.07). According to the principle of parsimony, a one-factor structure is preferred over a four-factor structure for the P-MoVac-COVID19S-9.

This study examined the concurrent validity of both the P-MoVac-COVID19S-12 and P-MoVac-COVID19S-9 by using Spearman’s rank correlation coefficient with three external criteria (Table 5). The results indicated significant correlations between the questionnaire scores and the number of vaccine doses received by parents (*r* = 0.209 (12-item version); *r* = 0.164 (9-item version)), the intention of parents to allow their children to be vaccinated (*r* = 0.685 (12-item version); *r* = 0.610 (9-item version)), and the level of worry about the adverse effects of vaccination on children (*r* = –0.361 (12-item version); *r* = –0.243 (9-item version); all *p* < 0.05). The two versions of the P-MoVac-COVID19S had a high correlation (*r* = 0.909). Additionally, both versions of P-MoVac-COVID19S had good internal consistency (α = 0.860 (12-item version); α = 0.897 (9-item version)) and test–retest reliability (*r* = 0.780 (12-item version); *r* = 0.652 (9-item version)).

## 4. Discussion

The present study adapted the MoVac-COVID19S to create a parent version of the scale (i.e., the P-MoVac-COVID19S) for assessing parents’ motivation to vaccinate their children against COVID-19. The factor structure of the 12-item P-MoVac-COVID19S aligned well with the theoretical framework of the cognitive model of empowerment [32]. Although we suggested that the one-factor structure was preferred over the four-factor structure for the 9-item P-MoVac-COVID19S based on the principle of parsimony, the *χ*^2^ difference between the P-MoVac-COVID19S-12 and P-MoVac-COVID19S-9 was nonsignificant. Therefore, this study supported the psychometric propensities of the P-MoVac-COVID19S based on both theoretical and empirical evidence. Moreover, validating the efficacy of the P-MoVac-COVID19S is crucial because studies have revealed that parents’ vaccine hesitancy affects children’s COVID-19 vaccine uptake [6,7,8,9,10,11,12,13,14].

Adopting the aforementioned theoretical framework [32] contributed to the psychometric soundness of the P-MoVac-COVID19S with respect to its factor structure, internal consistency, test–retest reliability, and concurrent validity. Good internal consistency indicates that the P-MoVac-COVID19S items measure the same concept of vaccine uptake motivations for parents in a consistent manner. Satisfactory test–retest reliability indicates that the P-MoVac-COVID19S assesses the concept of vaccine uptake motivation consistently over time. In addition, the P-MoVac-COVID19S had acceptable concurrent validity, as indicated by its significant associations with the number of vaccine doses received by parents and the level of worry regarding the adverse effects of vaccination in children. Parents who received a higher number of vaccine doses were more accepting of vaccination against COVID-19, which predicted a higher level of parental motivation for vaccinating children. By contrast, higher levels of worry about the adverse effects of vaccination in children may increase parents’ vaccine hesitancy and reduce parents’ motivation to have their child vaccinated. These associations support the concurrent validity of the P-MoVac-COVID19S and align with protection motivation theory [35,36,37,38,39]. Therefore, this study verified the utility of the P-MoVac-COVIDS for assessing parents’ motivation to have their child vaccinated against COVID-19.

There have been several instruments used for measuring parental attitudes about child COVID-19 vaccines. For example, the Parent Attitudes about Childhood Vaccines (PACV) survey [46] is a valid tool that has been successfully used to delineate the parental vaccine hesitancy before the COVID-19 pandemic. The 15-item [47] and 4-item PACV surveys [48] have been validated to be used in measuring parental attitudes and beliefs about childhood vaccines for COVID-19. The 15-item PACV survey contains three factors, including attitude, safety and efficacy, and analyzes behavior based on concepts that were developed based on the Health Belief Model [49]. The WHO’s Vaccine Hesitancy Scale has been also used to assess parental attitudes about childhood vaccines against COVID-19 [50,51]; however, its psychometric propensities in measuring parental attitudes about childhood vaccines against COVID-19 have not been examined. The P-MoVac-COVID19S has cognitive constructs similar to the original MoVac-COVID19S; therefore, parents’ motivations to vaccinate themselves and their children can be compared.

Several implications can be noted based on the findings of this study. Although many studies have reported a strong intention of parents to have their child vaccinated against COVID-19, many parents still have a low acceptance of vaccination [8,10,12]. Given that the P-MoVac-COVID19S contains the components of values, impacts, knowledge, and autonomy regarding children’s vaccination against COVID-19, healthcare providers and researchers can employ the P-MoVac-COVID19S to comprehensively analyze the multi-dimensional attitudes toward the vaccination of children and the underlying factors affecting parents’ unwillingness to vaccinate their children. It is also needed to investigate individual and environmental factors that could influence parents’ motivation to vaccinate their children, especially the attitudes of healthcare providers and trust in the healthcare system. Intervention programs must be developed based on the results regarding parental attitudes to address parents’ low acceptance of their child’s vaccination. The findings of the four-factor structure of the P-MoVac-COVID19S-12 highlight the value of empowering parental cognition regarding child COVID-19 vaccinations, which can equip medical professionals with a deeper understanding of parents’ hesitancy to vaccinate their children. Moreover, discrepancies in parents’ motivations to vaccinate themselves and their children and related factors should be emphasized. Given that the psychometric propensities of the MoVac-COVID19S have been validated in populations of various regions, further studies are needed to examine the psychometric propensity of the P-MoVac-COVID19S in populations of various regions and compare the levels of motivation and related factors.

This study has several limitations. First, we collected data from parents but not other informants; this could result in bias from shared-method variances [52]. Participants might also give socially desirable responses instead of choosing responses that are reflective of their true feelings. Collecting information regarding how many doses of COVID-19 vaccines parents and children have actually been administered might help reduce social desirability bias. Moreover, the participants were parents who were interested in the study purpose; therefore, the recruited participants were likely to be parents with a certain level of concern for the wellbeing of their children. Second, the participants were recruited using convenience sampling, which restricts the representativeness of the sample. Although recruiting participants using the online advertisement can deliver large numbers of participants quickly [53], Internet users may not be representative of the population. For example, a review study reported that recruiting participants using Facebook might have a bias in favor of young adults and people with higher education and incomes [54]. Future studies are thus required to enhance the representativeness of the sample and to corroborate the present findings. Third, several potential factors associated with parental willingness to allow their children to be vaccinated were not evaluated. For example, the attitudes of healthcare providers toward COVID-19 vaccines [55,56] and trust in the healthcare system [57] are potential factors associated with individuals’ attitudes toward vaccination. Therefore, future studies should incorporate additional relevant factors to reevaluate the psychometric soundness of the P-MoVac-COVID19S.

## 5. Conclusions

The present study demonstrated that the P-MoVac-COVID19S is a reliable and valid instrument for assessing parents’ motivation to have their child vaccinated against COVID-19. The P-MoVac-COVID19S incorporates four cognitive traits from the cognitive model of empowerment that help healthcare providers obtain information about parents’ willingness to vaccinate their children. Studies have documented the benefits of COVID-19 vaccines [58], and healthcare providers and relevant stakeholders hope to increase the vaccination rate. Therefore, healthcare providers could employ the P-MoVac-COVID19S to understand parents’ concerns about their children’s vaccination and to develop programs to improve parents’ willingness to vaccinate their children.

## Figures and Tables

**Table 1 vaccines-11-01192-t001:** Common instruments for assessing individuals’ attitudes toward COVID-19 vaccines.

Instrument	Authors	Number of Items	Content	Tested Region
Arizona CoVHORT Vaccine Questionnaire	Habila et al., 2022 [18]	10	Perceptions and beliefs regarding COVID-19 vaccines	Arizona, USA
Adult Vaccine Hesitancy Scale	Akel et al., 2021 [19]	10	Vaccine hesitancy	China and USA
Oxford COVID-19 Vaccine Hesitancy Scale	Freeman et al., 2022 [20]	7	Vaccine hesitancy	UK
COVID-19 Vaccine Attitudes and Beliefs Scale	Huang et al., 2022 [21]	15	Safety, efficacy, and general attitudes	China
COVID-19 Vaccine Concerns Scale	Gregory et al., 2022 [22]	7	Vaccine hesitancy	USA
Multidimensional COVID-19 Vaccine Hesitancy Scale	Kotta et al., 2022 [23]	15	Skepticism, risk, and fear of vaccines	Romania
COVID-19 Vaccine Hesitancy Questionnaire	Cvjetković et al., 2022 [24]	8	Confidence, complacency, and convenience of vaccines	Serbia
Vaccine Hesitancy Scale on Knowledge, Attitude, Trust and Vaccination Environment	Zhao et al., 2022 [25]	30	Vaccine hesitancy in knowledge, attitude, trust, and environment domains	China
COVID-19 Vaccine Hesitancy Scale in Qatar	Hammoud et al., 2023 [26]	50	Vaccine hesitancy, COVID-19 perceived risk, conspiracy beliefs, vaccine confidence, medical mistrust, and vaccine literacy	Qatar
5C Scale	Abd ElHafeez et al., 2021 [27]	10	Confidence, complacency, constraints, calculation, and collective responsibility	Middle-Eastern countries
Motors of COVID-19 Vaccination Acceptance Scale	Chen et al., 2021 [28] Fan et al., 2022 [29] Yeh et al., 2021 [30]	9-item and 12-item versions	12-item version: values, impacts, knowledge, and autonomy	Taiwan, mainland China, India, Ghana, Afghanistan, Indonesia, and Malaysia

**Table 2 vaccines-11-01192-t002:** Participants’ characteristics (*N* = 550).

	M (SD)	*n* (%)
Parent age (year)	44.29 (5.24)	
Child age (year)	11.80 (3.57)	
Parent sex		
Male		123 (22.4)
Female		427 (77.6)
Child sex		
Male		301 (54.7)
Female		249 (45.3)
Number of years parents received education	16.25 (2.47)	
Number of vaccine jabs in parents		
0		4 (0.7)
1		6 (1.1)
2		27 (4.9)
3		287 (52.2)
4		191 (34.7)
5		34 (6.2)
6		1 (0.2)
Intention to let children be vaccinated (1–10 VAS scale)		7.87 (2.04)
Worry about the adverse effects of child vaccination		
Not at all worried		23 (4.2)
Slightly worried		254 (46.2)
Moderately worried		144 (26.2)
Very worried		78 (14.2)
Extremely worried		51 (9.3)

**Table 3 vaccines-11-01192-t003:** Item properties of the P-MoVac-COVID19S.

	Mean (SD)				*n* (%)				Test–Retest
		1	2	3	4	5	6	7	
Item 1	5.25 (1.39)	15 (2.7)	15 (2.7)	34 (6.2)	50 (9.1)	166 (30.2)	182 (33.1)	88 (16.0)	0.67
Item 2	5.38 (1.25)	8 (1.5)	6 (1.1)	29 (5.3)	62 (11.3)	164 (29.8)	183 (33.3)	98 (17.8)	0.56
Item 3	5.83 (1.12)	6 (1.1)	3 (0.5)	13 (2.4)	27 (4.9)	123 (22.4)	214 (38.9)	164 (29.8)	0.39
Item 4	5.41 (1.36)	9 (1.6)	14 (2.5)	29 (5.3)	57 (10.4)	146 (26.5)	174 (31.6)	121 (22.0)	0.65
Item 5	5.53 (1.06)	0 (0.0)	6 (1.1)	19 (3.5)	49 (8.9)	176 (32.0)	201 (36.5)	99 (18.0)	0.50
Item 6	5.74 (1.02)	3 (0.5)	1 (0.2)	10 (1.8)	38 (6.9)	145 (26.4)	227 (41.3)	126 (22.9)	0.54
Item 7	3.89 (1.76)	42 (7.6)	77 (14.0)	156 (28.4)	92 (16.7)	53 (9.6)	70 (12.7)	60 (10.9)	0.43
Item 8	5.28 (1.08)	2 (0.4)	5 (0.9)	19 (3.5)	99 (18.0)	172 (31.3)	192 (34.9)	61 (11.1)	0.62
Item 9	5.86 (1.17)	9 (1.6)	5 (0.9)	7 (1.3)	33 (6.0)	97 (17.6)	228 (41.5)	171 (31.1)	0.47
Item 10	3.74 (1.63)	42 (7.6)	92 (16.7)	128 (23.3)	123 (22.4)	70 (12.7)	60 (10.9)	35 (6.4)	0.64
Item 11	4.28 (1.79)	26 (4.7)	82 (14.9)	104 (18.9)	81 (14.7)	92 (16.7)	88 (16.0)	77 (14.0)	0.42
Item 12	5.20 (1.21)	7 (1.3)	8 (1.5)	20 (3.6)	109 (19.8)	166 (30.2)	169 (30.7)	71 (12.9)	0.46

Note: Items 7, 10, and 11 are negatively worded items with reverse coding. There were 50 parents who completed the retest. Test–retest reliability was assessed using Spearman’s rho. P-MoVac-COVID19S: Parent version of Motors of COVID-19 Vaccination Acceptance Scale.

**Table 4 vaccines-11-01192-t004:** Factor structures of the 12-item and 9-item P-MoVac-COVID19Ss.

	12-Item MoVac-COVID19S		9-Item MoVac-COVID19S	
	Four-Factor	One-Factor	Four-Factor	One-Factor
Scale properties				
χ^2^ (df)	235.83 (48)	283.14 (54)	29.89 (22)	40.22 (27)
*p*-value	<0.001	<0.001	0.12	0.049
CFI	0.945	0.933	0.997	0.995
TLI	0.924	0.918	0.995	0.993
RMSEA	0.084	0.088	0.026	0.030
90% CI RMSEA	0.074, 0.095	0.078, 0.098	0.000, 0.047	0.002, 0.048
SRMR	0.088	0.095	0.050	0.059
Item factor loadings				
Item 1	0.725	0.690	0.749	0.734
Item 2	0.751	0.742	0.861	0.777
Item 3	0.828	0.802	0.836	0.808
Item 4	0.786	0.748	0.804	0.788
Item 5	0.685	0.676	0.740	0.674
Item 6	0.892	0.861	0.892	0.862
Item 7	0.388	0.229	-	-
Item 8	0.800	0.775	0.794	0.770
Item 9	0.531	0.388	1.000	0.369
Item 10	0.493	0.474	-	-
Item 11	0.604	0.366	-	-
Items 12	0.582	0.572	0.555	0.550

Note: Items 7, 10, and 11 are negatively worded items with reverse coding. In four-factor structures, items 3, 6, and 8 are embedded in the values construct; items 1, 4, and 12 are embedded in the impacts construct; items 2, 5, and 10 are embedded in the knowledge construct; and items 7, 9, and 11 are embedded in the autonomy construct. P-MoVac-COVID19S: Parent version of Motors of COVID-19 Vaccination Acceptance Scale.

**Table 5 vaccines-11-01192-t005:** Concurrent validity (Spearman’s rho) of the 12-item and 9-item P-MoVac-COVID19Ss.

	12-Item P-MoVac-COVID19S	9-Item P-MoVac-COVID19S
12-item MoVac-COVID19S	-	0.909
Number of vaccine jabs in parents	0.209	0.164
Intention to let children be vaccinated	0.685	0.610
Worry about the adverse effects child vaccination	−0.361	−0.243

Cronbach’s α = 0.860 (12-item P-MoVac-COVID19S) and 0.897 (9-item P-MoVac-COVID19S). For test–retest reliability, Spearman’s rho = 0.780 (12-item P-MoVac-COVID19S) and 0.652 (9-item P-MoVac-COVID19S). Note: all *p*-values < 0.001.

## Data Availability

The data will be available upon reasonable request to the corresponding authors.

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
