# Peer review of "A Parent Version of the Motors of COVID-19 Vaccination Acceptance Scale for Assessing Parents’ Motivation to Have Their Children Vaccinated"

_vaccines, 2023, doi:10.3390/vaccines11071192_

Round 1
Reviewer 1 Report
This is a nice article with lots of innovative ideas. I have a few minor comments.
1. Please state the research questions or hypotheses that the study aims to address.
2. Please include more information about the process of adapting the original MoVac-COVID19S to create the parent version, such as any modifications made to the wording or content of the items.
3. Please provide more details about the time interval between the test and retest and the criteria used to determine satisfactory reliability.
4. Please explain the use of diagonally weighted least squares as the estimation method for the CFA.
5. Please mention the rationale for including these specific demographic variables and how they relate to the research questions or objectives.
6. Please note the limitation of self-report measures from parents, introducing the potential for single-rater bias and social desirability bias.
7. Please discuss that convenience sampling was used, which may limit the generalizability of the results, lacking more diverse and representative samples.
8. Is it possible to investigate certain factors that could influence parental willingness to vaccinate their children, such as the attitudes of healthcare providers and trust in the healthcare system?
Author Response
We appreciated your valuable comment. As discussed below, we have revised our manuscript with underlines based on your suggestions. Please let us know if we need to provide anything else regarding this revision.
Comment
- Please state the research questions or hypotheses that the study aims to address.
Response
Thank you for your comment. We added research questions and hypotheses as below. Please refer to line 117-129.
“The present study aimed to create the parent version of the MoVac-COVID19S (P-MoVac-COVID19S) by modifying the MoVac-COVID19S and examine its psychometric propensities, including factor structures, test–retest reliability, and internal consistency of the 9-item and 12-item P-MoVac-COVID19S. We hypothesized that the 12-item P-MoVac-COVID19S had a four-facto structure (values, impacts, knowledge, and autonomy) and the 9-item P-MoVac-COVID19S had one-facto structure corresponded well with CME theory. We hypothesized that the 9-item and 12-item P-MoVac-COVID19S had acceptable internal consistency, test–retest reliability. According to protection motivation theory [35–39], we also hypothesized that the 9-item and 12-item P-MoVac-COVID19S had acceptable concurrent validity by analyzing the association between parental motivation to vaccinate their children and the level of parental worry regarding the adverse effects of COVID-19 vaccination on children’s health and the number of COVID-19 vaccine doses received by parents.”
Comment
- Please include more information about the process of adapting the original MoVac-COVID19S to create the parent version, such as any modifications made to the wording or content of the items.
Response
We added the process of adapting the original MoVac-COVID19S to create the parent version as below. Please refer to line 150-155.
“The research team adapted the MoVac-COVID19S into P-MoVac-COVID19S by replacing the words ‘‘me” and “my” by “my child” and “my child’s,” respectively. The item 7 “I feel pressured about receiving COVID-19 vaccine” was transformed into “I feel pressured about letting my child receive COVID-19 vaccine.” The item 11 “I only receive COVID-19 vaccine if it is required” was transformed into “I only let my child receive COVID-19 vaccine if it is required.””
Comment
- Please provide more details about the time interval between the test and retest and the criteria used to determine satisfactory reliability.
Response
We have now provided the information regarding the time interval between the test and retest assessments. Regarding the criterion used to determine satisfactory test-retest reliability, we used an r > 0.4 indicating satisfactory. The information is now clearly mentioned in the revised manuscript. Please refer to line 187-191.
“Spearman’s rank correlation coefficients were employed to evaluate the test–retest reliability (with a time interval of one week) of each item on the P-MoVac-COVID19S and of the two summed scores (i.e., the 9-item and 12-item P-MoVac-COVID19S). When the correlation coefficient is larger than 0.5, the test-retest reliability is considered to be satisfactory; when the coefficient is between 0.3 and 0.5, the reliability is acceptable [42].”
Comment
- Please explain the use of diagonally weighted least squares as the estimation method for the CFA.
Response
We have now explained why using the diagonally weighted least squares as the estimation method. Please refer to line 195-197.
“The diagonally weighted least squares estimator was used because it can handle the ordinal scale such as the Likert-type scales used in the P-MoVac-COVID19S [43].”
Comment
- Please mention the rationale for including these specific demographic variables and how they relate to the research questions or objectives.
Response
We added the rationale for including specific demographic variables as below. Please refer to line 179-183.
“Studies have found that parental motivation of vaccinating their child varied across different sex and ages of parents and children [13,40,41]; therefore, data on the sex and ages of the parents and their children were collected. Moreover, parental education level was a key factor influencing pediatric COVID-19 vaccine hesitancy [40]; therefore, the parents were asked to state how many years of education they had received.”
Comment
- Please note the limitation of self-report measures from parents, introducing the potential for single-rater bias and social desirability bias.
Response
We added the explanations for the limitation of self-report measures from parents as below. Please refer to line 323-329.
“First, we collected data from parents. The use of only a single data source may result in shared-method variances [52]. Participants might also give socially desirable responses instead of choosing responses that are reflective of their true feelings. Collecting information regarding how many doses of COVID-19 vaccines parents and children have actually been administered might help reduce social desirability bias.”
Comment
- Please discuss that convenience sampling was used, which may limit the generalizability of the results, lacking more diverse and representative samples.
Response
We added the discussion about the limitations resulted from convenience sampling as below. Please refer to line 332-336.
“Although recruiting participants using the online advertisement can deliver large numbers of participants quickly [53], Internet users may not be representative of the population. For example, a review study reported that recruiting participants through Facebook might have a bias in favor of young adults and people with higher education and incomes [54].”
Comment
- Is it possible to investigate certain factors that could influence parental willingness to vaccinate their children, such as the attitudes of healthcare providers and trust in the healthcare system?
Response
Thank you for your suggestion. We added it as one of the issues warranted further study as below. Please refer to line 310-312.
“It is also needed to investigate individual and environmental factors that could influence parental motivation to vaccinate their children, especially the attitudes of healthcare providers and trust in the healthcare system.”
Reviewer 2 Report
Thank you for the opportunity to review this study which reports the process of adaptation of the MoVac-COVID19S to create a parent version (P-MoVac-COVID19S) and reports the results of the psychometric soundness of this new tool. The topic is of interest and the study is well-organised overall. Here are my observation to the specific sections.
Introduction: The introduction is articulated quite well and contains an appropriate number of references. I would have given more space to literature surveys on vaccination existaction in parents, associated factors, differences between different countries. I would also have dealt more extensively with the instruments for the assessment of attitudes to vaccination: the study mentions only one, which is then chosen to be adapted for parents, but does not highlight whether there are others, and what characteristics of the one chosen, compared to the others, led the authors to prefer it. The first part on the importance of prevention is a bit redundant; these are now widely shared concepts.
Materials and Methods are presented clearly, but but I have a doubt about the number of participants: how many have declared themselves interested and how many have then made the appointment? Same applies to the Results section.
Discussion: I appreciated that the authors reported the limitations of the study. I would have wished for a more developed discussion though, for example expanding the implications and moving them before the limitations in the text. Also please cite if there are similar tools and which are the peculiarities of the proposed one compared to others, if applicable.
Moderate editing of English language required.
Author Response
We appreciated your valuable comment. As discussed below, we have revised our manuscript with underlines based on your suggestions. Please let us know if we need to provide anything else regarding this revision.
Comment 1
Introduction: The introduction is articulated quite well and contains an appropriate number of references. I would have given more space to literature surveys on vaccination existaction in parents, associated factors, differences between different countries.
Response
Thank you for your comment. We added a new paragraph to introduce them as below. Please refer to line 58-73.
“A systemic review and meta-analysis on 44 studies published between 2020 and 2021 found that the overall proportion of parents that intend to vaccinate their children against the COVID-19 was 60.1%, while the proportion of parents that refuse to vaccinate their children was 22.9% and the proportion of unsure parents was 25.8%; the main predictors of parents' intention to vaccinate their children were fathers, older age of parents, higher income, higher levels of perceived threat from the COVID-19, and positive attitudes towards vaccination [15]. Another review study published in 2021 also revealed that the factors influencing parents' attitudes towards child vaccination were heterogeneous, reflecting country-specific factors, but also displaying some similar trends across countries, such as the education level of parents; the leading reason in the child vaccination decision was to protect children, family and others; and the fear of side effects and safety was the most important reason in not vaccinating children [16]. A review study on 118 surveys representing 55 different countries demonstrated that the willingness of parents to vaccinate their children were influenced by public attitudes; a more positive public attitude towards vaccination correlated with higher vaccination rates [17].”
Comment 2
I would also have dealt more extensively with the instruments for the assessment of attitudes to vaccination: the study mentions only one, which is then chosen to be adapted for parents, but does not highlight whether there are others, and what characteristics of the one chosen, compared to the others, led the authors to prefer it.
Response
Thank you for your comment. We added a new paragraph to introduce the instruments used to measure the attitudes toward COVID-19 vaccines as below. We also summarized them in Table 1. Please refer to line 77-111.
“There have been several instruments developed for measuring individuals’ attitudes toward COVID-19 vaccines such as the Arizona CoVHORT Vaccine Questionnaire [18], Adult Vaccine Hesitancy Scale [19], Oxford COVID-19 Vaccine Hesitancy Scale [20], COVID-19 Vaccine Attitudes and Beliefs Scale [21], COVID-19 Vaccine Concerns Scale [22], Multidimensional COVID-19 Vaccine Hesitancy Scale [23], COVID-19 Vaccine Hesitancy Questionnaire [24], Vaccine Hesitancy Scale on Knowledge, Attitude, Trust and Vaccination Environment [25], COVID-19 Vaccine Hesitancy Scale in Qatar [26], and 5C Scale [27]. The Motors of COVID-19 Vaccination Acceptance Scale (MoVac-COVID19S) is a valuable tool for assessing individuals’ attitudes and factors affecting their decision to be vaccinated against COVID-19 [28–30]. The MoVac-COVID19S has several advantages in measuring individuals’ attitudes toward COVID-19 vaccines. First, the MoVac-COVID19S was adapted from the Motors of Influenza Vaccination Acceptance Scale, which was developed for assessing acceptance of the influenza vaccine [31]. The MoVac-COVID19S, which is based on the cognitive model of empowerment (CME) [32], incorporates four core cognitive components that determine individuals’ motivation to be vaccinated against COVID-19, namely values (i.e., how much an individual cares about the purpose of vaccination), impacts (i.e., how much an individual believes in the effectiveness of COVID-19 vaccination), knowledge (i.e., an individual’s level of knowledge regarding vaccination against COVID-19), and autonomy (i.e., an individual’s confidence and control over their decision to receive the COVID-19 vaccine). Compared with most of other instruments, the MoVac-COVID19S assesses a broader scope of understanding the attitudes toward COVID-19 vaccines. Second, most of other instruments have been validated among people living in a single country or region. Studies have verified that MoVac-COVID19S has acceptable psychometric soundness for assessing individuals’ motivation to be vaccinated in different populations, including in Taiwan, mainland China, India, Ghana, Afghanistan, Indonesia, and Malaysia [33,34]. Third, two versions of the MoVac-COVID19S have been proposed: a 9-item MoVac-COVID19S with all items worded positively and a 12-item MoVac-COVID19S with nine items worded positively and three items worded negatively [28]. The two versions of the MoVac-COVID19S offer a more flexible way of assessing individuals’ attitudes toward COVID-19 vaccines depending on the needs of the surveys. The summary of the instruments developed for assessing individuals’ attitudes toward being vaccinated against COVID-19 is shown in Table 1.”
Comment 3
The first part on the importance of prevention is a bit redundant; these are now widely shared concepts.
Response
We deleted the first eight lines in the original manuscript. Please refer to line 48.
Comment 4
Materials and Methods are presented clearly, but I have a doubt about the number of participants: how many have declared themselves interested and how many have then made the appointment? Same applies to the Results section.
Response
We added the explanation for them as below. Please refer to line 139-145.
“In total, 562 parents expressed interest in participating in research; of them, 12 parents were excluded because of their children’s age (younger than 6 or older than 18 years). A total of 550 parents participated in the study. The research assistants evaluated potential participants in an on-site study room to determine whether they had signs of impaired intellect or substance use that might interfere with their understanding of the study’s purpose or prevent them from completing the questionnaire. No participants were excluded.”
Comment 5
Discussion: I appreciated that the authors reported the limitations of the study. I would have wished for a more developed discussion though, for example expanding the implications and moving them before the limitations in the text.
Response
Thank you for your suggestion. We expanded the implications as below and moved them before the limitations in the text. Please refer to line 303-323.
“Several implications can be made based on the findings of this study. Although many studies have reported a strong willingness of parents to vaccinate their children against COVID-19, many parents still have a low acceptance of vaccination [8,10,12]. Given that the P-MoVac-COVID19S contains the components of values, impacts, knowledge, and autonomy regarding children’s vaccination against COVID-19, health-care providers and researchers can employ the P-MoVac-COVID19S to comprehensively analyze the multi-dimensional attitudes toward vaccination of their children and underlying factors affecting parental unwilling to vaccinate their children. It is also needed to investigate individual and environmental factors that could influence parental motivation to vaccinate their children, especially the attitudes of healthcare providers and trust in the healthcare system. Appropriate programs may then be designed based on this information specific to the components of parental attitudes to address low parental acceptance of COVID-19 vaccination uptake for their children. The factor structure findings related to the P-MoVac-COVID19S highlight the value of cognitive empowerment, which can equip health-care providers and research personnel with a deeper understanding of parental hesitancy to vaccinate their children. Moreover, discrepancies in parents’ motivations to vaccinate themselves and their children and related factors should emphasize. Given that the psychometric propensities of the MoVac-COVID19S have been validated in the populations of various regions, further study is needed to examine the psychometric propensity of the P-MoVac-COVID19S in the populations of various regions and compare the levels of motivation and related factors.”
Comment 6
Also please cite if there are similar tools and which are the peculiarities of the proposed one compared to others, if applicable.
Response
Thank you for your comment. We added a new paragraph to discuss the instruments used to measure parental attitudes toward vaccinating children against COVID-19 as below. Please refer to line 290-302.
“There have been several instruments used for measuring parental attitudes about COVID-19 vaccines for children. For example, the Parent Attitudes about Childhood Vaccines (PACV) [46] is a valid tool that has been successfully used to delineate the parental vaccine hesitancy before the COVID-19 pandemic. The 15-item [47] and 4-item PACV [48] have been validated to be used in measuring parental attitudes and beliefs about childhood vaccines for COVID-19. The 15-item PACV contained three factors, including attitude, safety and efficacy, and behavior based on the concepts of developed based on the Health Belief Model [49]. The WHO’s Vaccine Hesitancy Scale has been also used to assess parental attitudes about childhood vaccines against COVID-19 [50,51]; however, its psychometric propensities in measuring parental attitudes about childhood vaccines against COVID-19 have not been examined. The P-MoVac-COVID19S has the cognitive constructs similar to the original MoVac-COVID19S; therefore, parents’ motivations to vaccinate themselves and their children can be compared.”